# Bridging the Gap between Theoretical Learning and Practical Application: A Qualitative Study in the Italian Educational Context

Stefania Fantinelli [1], Michela Cortini [2], Teresa Di Fiore [2], Stefano Iervese [3] and Teresa Galanti [2,*]

[1] Department of Human Studies, University of Foggia, 71121 Foggia, Italy
[2] Department of Psychological, Health and Territorial Sciences, D'Annunzio University of Chieti-Pescara, 66100 Chieti, Italy
[3] Department of Human Resource, Mylia Advancing Humanity, 66100 Chieti, Italy
* Correspondence: teresa.galanti@unich.it

**Abstract:** In the contemporary educational landscape, there is a growing recognition of the transformative impact of practical experiences within traditional learning frameworks. This shift reflects a pedagogical evolution that values contextualized learning and the acquisition of practical skills together with theoretical knowledge. In the Italian educational context, School–Work Alternation (SWA) represents a proactive response to the evolving needs of the workforce and the imperative for educational institutions to prepare students for professional life. This study's objectives include a deep exploration of students' SWA experience, evaluating its impact on employability perceptions and the sense of agency, examining the influence of Self-Orientation, and contributing insights to the discourse on integrating practical experiences in education. Employing a mixed methodology and a bottom-up approach, 63 high school students of different Italian regions participated in an online in-depth interview and an ad hoc questionnaire designed to measure the experience of SWA in relation to variables of interest, utility, advantage, perception of support, quality of received mentoring, engagement, and satisfaction. The quantitative results indicate that personal choice significantly influences the perceived usefulness of and satisfaction with SWA, with those students guided by curiosity exhibiting higher utility and satisfaction. Qualitative analysis underscores both positive and negative aspects, with respondents viewing SWA as a useful experience bridging work, corporate, and school realms provided that students are key players in the SWA experience's choice and that the SWA's partners are motivated to guide them in this practical training. Moreover, results highlight SWA's relevance in guiding academic and career paths, emphasizing its potential to offer valuable support to students. This study contributes nuanced insights into integrating practical experiences in education, offering recommendations for educators and policymakers to better prepare students for the dynamic demands of the modern job market.

**Keywords:** theoretical and practical learning; workplace transition; agency; employability; soft skills



## 1. Introduction

In the contemporary educational landscape, there is a growing recognition of the transformative impact that practical experiences can exert when integrated into traditional learning frameworks [1–3]. This paradigm shift reflects a pedagogical evolution that acknowledges the efficacy of contextualized learning and the acquisition of practical skills in tandem with theoretical knowledge [4–6]. In the Italian context, one notable manifestation of this educational evolution is the School–Work Alternation (SWA) program. SWA represents an innovative initiative designed to bridge the gap between classroom learning and real-world application [7]. This program aims to provide students with firsthand exposure to authentic work environments, fostering a symbiotic relationship between academic knowledge and practical skills acquisition [8]. By allowing students to rotate between

school-based instruction and workplace immersion, SWA serves as an example of how educational institutions are adapting to the demands of the contemporary job market and preparing students for the complexities of professional life [9–11]. As educators and policymakers grapple with the need to equip students with not only theoretical knowledge but also practical competencies, SWA emerges as a compelling model that warrants in-depth exploration. So, this study contributes to scrutinize the lived experiences of high school Italian students participating in SWA, employing a qualitative methodology and adopting a bottom-up approach. The research also aims to shed light on the nuanced dynamics of SWA, its impact on student's perceptions of employability, and its role in shaping their sense of agency and self-orientation. Through an examination of SWA's dimensions, this study seeks to contribute valuable insights into the ongoing discourse on the integration of practical experiences in contemporary education, as well as the crucial role of soft skills, increasingly in demand in current work environments.

## 2. Background

### 2.1. Overview of School to Work Alternation (SWA)

School–Work Alternation (SWA) in the Italian educational context represents a proactive response to the evolving needs of the workforce and the imperative for educational institutions to prepare students for professional life. SWA typically starts during the last three years of upper secondary school, around ages 15 to 16, depending on the specific school and program. The Italian educational system is divided into several stages. It begins with the first voluntary stage for children aged 3 to 6. Following this, there is the first compulsory grade known as primary school, which caters to students from ages 6 to 11. Lower secondary school is also compulsory and lasts for three years. Subsequently, there is an additional stage: upper secondary school, which is not compulsory and consists of several tracks. High school (or lyceum) can be focused on various subjects such as humanities, sciences, languages, and arts. Technical institutes offer a more vocational education with a focus on technical subjects and professional skills. Professional institutes provide vocational training for various professions, including technical, commercial, and social services.

SWA is deeply rooted in the broader European tradition of work-based learning, where initiatives like apprenticeships have been recognized for their efficacy in cultivating practical skills alongside academic knowledge [12,13]. In Italy, SWA is embedded within the national educational system and regulated by the Ministry of Education, Universities and Research (MIUR). The program typically involves students alternating between traditional classroom settings and real-world workplaces, providing them with firsthand exposure to the demands and dynamics of various professions (Law no 53/2003, legislative decree 77/2005). Recently, Law n. 107/2015 has reiterated the importance of pairing theoretical knowledge with practical skills, strengthening the school's relationships with the local community, the productive world, and the service sector.

The regulations regarding SWA in Italy can vary depending on the region and school. However, the MIUR has provided some general guidelines explaining the overall requirements that companies should meet to host students in SWA programs. For example, common requirements include the company designating an internal tutor responsible for guiding and supporting students during the SWA period, providing guidance and supervision. The company must be able to offer meaningful learning activities consistent with the educational objectives set by the school. A formal agreement, defining the details of the SWA arrangement, including obligations, rights, and responsibilities of the involved parties, must be established between the school and the company.

According to regulations, the number of hours that students can spend at a company varies based on the type of school: 90 h for high schools, 150 h for technical institutes, and 210 h for vocational institutes. The activity must be carried out at the same hosting company. The requirement for an agreement between the company and the school, along with the explicit definition of learning objectives, implies that employees should be informed and

prepared to welcome young students. Collaborations between schools and businesses can promote innovative teaching methods [14–16] and the dissemination of educational processes geared towards acquiring skills applicable in the workplace [7,17]. Concurrently, this fosters guidance, a culture of self-entrepreneurship [18,19], active citizenship [20], and increased engagement of young individuals in learning processes, facilitated by new technologies [21–23]. Research by Fettes and colleagues [24] stressed that it is not simply a matter of "skills transfer", from educational to work context, but a continuous and transformative process during which individuals learn how to recontextualize skills to suit different activities and environments. In this perspective, SWA could have a positive impact on students' skill development and employability, emphasizing the program's role in bridging the gap between theoretical learning and practical application before officially entering the workplace, with a focus on soft skills.

### 2.2. Theoretical Framework: Employability and Sense of Agency

The theoretical underpinnings of this study are anchored in the concepts of employability and agency. Employability, as defined by several scholars [25–27], is not merely the possession of job-specific skills but encompasses a broader set of attributes, including problem-solving abilities, interpersonal skills, and adaptability. It also refers to an individual's ability to enter the labor market on a stable basis, regardless of whether individuals find a job. The intertwining of these factors helps individuals to better adapt to the working environment, which is now changed and uncertain [28–31]. Furthermore, both internal and external variables, such as the context of life or work, converge in the construct. This interplay is underscored by a recent study [32], which highlights the crucial role of employability promotion in diversity management strategies to foster job inclusiveness. Moreover, employability also includes the dimensions of adaptability [33,34], career identity [35,36] and human and social capital [37–39]. Adaptability involves the ability to adapt to unexpected tasks and changes, including the propensity to learn, optimism, openness to change, and internal locus of control. The second dimension, career identity, explores individuals' self-perception and aspirations within the working context [37,40]. Lastly, the human and social capital dimensions encompass the utilization of social connection to enhance employment opportunities and the personal factors influencing job prospects and career progression, such as age, education, and work experience [41,42].

Furthermore, the concept of agency becomes integral to understanding the impact of SWA on students. Sense of agency refers to an individual's perception of their capacity to influence their own life course and the world around them [43]. It embodies a deep-seated belief in one's ability to actively shape and control the outcomes of one's actions. Several studies underlined that a robust sense of agency has far-reaching implication for an individual's employment, influencing both the mindset and actions in the pursuit of a successful career [44,45]. Firstly, individuals with a heightened sense of agency tend to approach challenges with a proactive attitude [46], viewing obstacles as opportunities for skill development and personal growth. Moreover, the proactive orientation instilled by a strong sense of agency is closely linked to increased motivation and engagement [47]. As individuals actively shape their learning experiences and career paths, they are more likely to persevere in the face of difficulties, demonstrating a resilience that is crucial in today's competitive job market [48,49]. This resilience become particularly evident during the period of unemployment or career transitions, where individuals with a robust sense of agency are more likely to approach these challenges as opportunities for reevaluation and strategic career planning.

### 2.3. Empowering Employability and Sense of Agency through School-Work Alternance

In the contemporary landscape, there is a growing imperative for educational programs and initiatives that cultivate a heightened awareness of employability and sense of agency among young individuals, while nurturing the development of versatile and orientation skills [50]. Within this context, School–Work Alternation (SWA) emerges as

a potential mechanism for enhancing both students' employability and sense of agency. By immersing students in varied work environments, SWA becomes a conduit for the cultivation of transferable skills [51–53].

For what concerns employability, SWA, by design, exposes students to practical, on-the-job scenarios, aligning theoretical learning with real-world application. The skills developed during SWA, such as adaptability, problem solving, and effective communication, are not only instrumental in immediate job roles but also contribute to the multidimensional construct of employability. So, School–Work Alternation (SWA) assumes a pivotal role in this regard, aiding students in skill development, career decision making, and self-orientation [54,55]. Through this methodology, a comprehensive formation of the individual is achievable, without undermining the indispensable role of culture [17,52].

Regarding the sense of agency, SWA could serve as a catalyst in nurturing students' sense of agency by providing immersive, real-world experiences. Through SWA, students encounter diverse challenges and responsibilities, allowing them to witness the tangible impact of their efforts and to cultivate resilience and adaptability, crucial components of a well-developed sense of agency. This active participation fosters a proactive mindset, encouraging students to see themselves as active contributors to their own educational and professional journey [43]. Moreover, empirical evidence from studies like [56,57] support the idea that work-integrated learning experiences contribute significantly to the development of a sense of agency among students. According to these studies, first hand exposure to the intricacies of professional life during SWA empowers students with the confidence to influence their trajectories, promoting a sense of agency that extends beyond the classroom.

## 3. Aim and Objectives

Starting from this theoretical framework, this study aims to comprehensively explore the lived experiences of high school students participating in the School–Work Alternation (SWA) program within the Italian educational context. The overarching goal is to gain insights into the multifaced impact of SWA on students' perceptions of employability, as well as its role in shaping their sense of agency and self-orientation.

To delve into the nuanced dynamics of SWA, this study adopted a qualitative methodology and a bottom-up approach to shed light on the School–Work Alternation experience and its implications. In particular, the following objectives were set:

1. Examine lived experiences, through an in-depth exploration of the experiences of high school students engaged in SWA, to understand the intricacies of their interactions with both working and educational settings.
2. Evaluate impact on perception of employability and sense of agency: we want to investigate how SWA influences student's perceptions of employability by analyzing the skills, attitudes, and insights they acquire during their participation in the program, as well as students' ability to exercise control over their learning and career path.
3. Examine the impact of self-orientation, investigating the influence of SWA on student's ability to actively navigate their career paths and their perception regarding the program's utility to assist them in their future vocational choice.
4. Contribute insights to educational discourse: this study aspires to contribute meaningful insights to the ongoing discourse surrounding the integration of practical experiences in contemporary education, providing recommendations for educators and policymakers to prepare students for the dynamic demands of the modern job market.

## 4. Materials and Methods

Following a qualitative explorative approach, an online in-depth interview was designed in order to collect data, giving space to the participants' viewpoints. The interview was preceded by an ad hoc questionnaire designed to measure the experience of School–Work Alternation in relation to variables of interest, utility, advantage, perception of support, quality of received mentoring, engagement, and satisfaction. Each of these aspects

was measured on a 5-point Likert scale. Table 1 provides detailed information on the variables considered and the anchors for each scale.

**Table 1.** Means, Standard Deviation, Skewness and Kurtosis of Study Variables.

| Variable | Mean | Standard Deviation | Skewness | Kurtosis | Likert Scale Anchors |
|---|---|---|---|---|---|
| Usefulness of SWA | 2.84 | 0.739 | 0.016 | −0.626 | From 1 (very little useful) to 5 (very much useful) |
| Interest in SWA | 2.76 | 0.729 | −0.365 | 0.169 | From 1 (very low interest) to 5 (very high interest) |
| Suitability of SWA environment | 3.21 | 0.881 | −0.306 | 0.005 | From 1 (very low suitability) to 5 (very high suitability) |
| Risk prevention in SWA | 2.72 | 1.06 | −0.065 | −0.661 | From 1 (very low prevention) to 5 (very high prevention) |
| Support in SWA | 3.15 | 0.754 | 0.410 | 0.135 | From 1 (very low support) to 5 (very high support) |
| SWA Tutorship | 3.29 | 0.764 | −0.341 | 0.383 | From 1 (very low availability) to 5 (very high availability) |
| Engagement in SWA | 3.32 | 0.692 | −390 | 0.236 | From 1 (zero engagement) to 5 (total engagement) |
| SWA Satisfaction | 2.86 | 0.737 | -0.255 | −0.074 | From 1 (not at all satisfied) to 5 (completely satisfied) |
| SWA Profitability | 2.60 | 0.896 | 0.336 | −0.211 | From 1 (not at all profitable to 5 (completely profitable) |

### 4.1. Data Collection

The structured interviews were distributed both through a link and a QR code, granting access to the Qualtrics online platform; there was a two-step recruitment process: firstly, we have recruited a convenience sample. Then, the recruited individuals were also requested to share the questionnaire access link with their schoolmates and peers at large, utilizing email and social media platforms (virtual snowball sampling). Ethical approval was considered unnecessary for this study, despite the inclusion of human participants. This determination was grounded in the absence of specialized procedures or treatments that might inflict stress or harm on the participants, eliminating ethical concerns. The research adheres to the principles outlined in the Declaration of Helsinki (World Medical Association Declaration of Helsinki: Ethical Principles for Medical Research Involving Human Subjects, 2008). Additionally, participants were duly informed about the handling of personal data, and we ensured anonymity in accordance with EU Regulation 2016/679.

The interview was created ad hoc according to the explorative aim of the study; some questions are reported as follows:

Do you think the School–Work Alternation experience has been useful for the future academic or professional choices you will have to make? Why?

Do you think this project can facilitate entry into the world of work? Why?

What are the most important things you have learned as professional skills?

### 4.2. Mix Method Data Analysis

To examine the lived experiences of the students in-depth and to understand the intricacies of their interactions with working, educational, and family settings, a mixed-method analysis was implemented, thus maintaining the rich insights associated with qualitative data while ensuring the precision characteristic of quantitative analysis methodologies. The primary advantage of the online mode lies in its ability to conserve resources and reach a broader, geographically dispersed sample, while also ensuring participant anonymity [58].

This anonymity is crucial as it enables respondents to provide honest answers, thereby minimizing the impact of social desirability [59].

For the quantitative data, a one-way ANOVA with the SPSS was performed to detect the opinion of the students about profitability and satisfaction of School–Work Alternance (SWA) experiences. This analysis intended to explore the difference between groups regarding the autonomy granted to the student in choosing the SWA activity, to highlight whether there were differences in terms of usefulness and satisfaction between those who based their choice on personal interests and curiosity and those who instead complied with the requests of the school or their family.

Concerning the qualitative data, various techniques were employed to analyze the qualitative and quantitative data to adopt a triangulation not only of the data collection methods but also of data analysis approaches [49,60]. This aligns with the Standards for Reporting Qualitative Research (SRQR) by [61] and the guidelines outlined by the Critical Appraisal Skills Program (2018) for qualitative research. Triangulation, as recommended by these standards, contributes to bolstering the trustworthiness and credibility of data analysis.

Qualitative analysis was conducted using a thematic approach [62], gaining insights into how students genuinely experienced their SWA, emphasizing their thoughts and beliefs expressed explicitly and implicitly. Before commencing the coding stage, three distinct researchers extensively reviewed the transcripts to gain a comprehensive understanding of the collected data and verify the accurate representation of participants' viewpoints.

The interviews underwent two coding steps: open coding involved assigning codes to meaningful words and sentences, while axial coding organized these codes into broader groups, which were subsequently integrated into overarching macro-categories during the selective coding phase. The careful choice of words by participants is particularly important, as often what individuals consider meaningful is conveyed through this linguistic form [63]. Various reasons make these data analysis methods especially fitting for our study. Firstly, given the study's explorative objective of offering a descriptive and interpretive overview of the collected data, thematic analysis proves valuable for identifying and examining themes and patterns within qualitative data [64]. Secondly, thematic analysis is well suited for interview and open-ended survey data. Each idea or concept expressed by participants corresponds to a unit of meaning; the minimum was considered at least one verb and one subject. The definitions of the codes were created by two researchers trained in qualitative analysis; then, the Cohen Kappa (0.85) has been calculated.

Quantitative analysis of the interviews was performed using T-LAB software [60]. We carried out an automatic analysis of the content to intercept the most frequently used words, and the occurrences and co-occurrences of words in the textual material. Specifically, the quantitative analysis conducted using T-LAB software allowed for the identification of word repetitions and the exploration of the most prevalent associations within the text.

*4.3. Participants and Procedures*

The sample comprised 63 high school students, encompassing various fields of study. The key criterion for inclusion in the study was the completion of SWA, ensuring that participants had practical work experience alongside their school pursuits. The age range of participants spanned from 16 to 19 years, with an average age of 17.95 years (standard deviation = 0.888). The gender distribution in the sample was quite unbalanced, as there were 18 males and 45 females. This diverse composition of participants, both in terms of school interests and gender, offers a comprehensive representation of high school students who have undergone SWA, providing a robust foundation for exploring the impacts and perspectives associated with this educational program. Regarding the SWA, only 46.1% of the sample expressed moderate to complete satisfaction with the lived experience. This noteworthy finding is further supported by the analysis of qualitative data, which reveals various challenges in the SWA experience, thoroughly examined in the discussion of results. Table 2 presents in more detail the socio-demographic characteristics of the sample.

**Table 2.** Socio-demographic characteristics of the sample.

| Variables | Categories | Numbers | Percentage | Average | Sd |
|---|---|---|---|---|---|
| 1. Gender | M | 18 | 28.6% | | |
| | F | 45 | 71.4% | | |
| | Other | | | | |
| | Total | 63 | 100% | | |
| 2. Age | 16 | 4 | 6.3% | | |
| | 17 | 15 | 23.8% | | |
| | 18 | 25 | 39.7% | | |
| | 19 | 19 | 30.2% | | |
| | Total | 63 | 100% | 17.94 | 0.896 |
| 3. School | High school | 49 | 77.8% | | |
| | Technical Institutes | 13 | 20.6% | | |
| | Professional Institutes | 1 | 1.6% | | |
| | Total | 63 | 100% | 1.23 | 0.465 |
| 4. SWA partner sector | Tourism | 10 | 15.9% | | |
| | Accounting | 4 | 6.3% | | |
| | Personal services | 22 | 34.9% | | |
| | Catering | 3 | 4.8% | | |
| | Education and training | 14 | 22.2% | | |
| | Transport | 1 | 1.6% | | |
| | Industry | 5 | 7.9% | | |
| | Publishing | 2 | 3.2% | | |
| | Entertainment | 1 | 1.6% | | |
| | Total | 63 | 100% | | |
| 5. Choice | Personal | 12 | 19% | | |
| | Familiar | 4 | 6.3% | | |
| | Scholastic | 37 | 58.7% | | |
| | Casual | 10 | 16.09% | | |
| | Total | 63 | 100% | | |
| 6. SWA Satisfaction | Not at all | 8 | 13.8% | 2.86 | 0.74 |
| | Slightly | 26 | 40% | | |
| | moderately | 21 | 32.3% | | |
| | Completely | 8 | 13.8% | | |
| | Total | 63 | 100% | | |
| 7. SWA Profitability | Not at all | 1 | 1.5% | 2.60 | 0.90 |
| | Slightly | 20 | 30.8% | | |
| | moderately | 30 | 47.7% | | |
| | Completely | 12 | 18.5% | | |
| | Total | 63 | 100% | | |

**5. Results**

*5.1. Quantitative Results*

Preliminary analyses revealed two particularly significant findings: firstly, the average scores on scales related to satisfaction and profitability of SWA were relatively low, with values of 2.86 and 2.60, respectively. Secondly, less than half of the participants considered themselves moderately or completely satisfied with SWA. These data indicate a general dissatisfaction and a limited perception of utility among participants, prompting the need for further investigation and analysis.

Consequently, we conducted two ANOVAs, to investigate differences in the perception of profitability and satisfaction towards the SWA between the group of students who had freely chosen the SWA's partner and the group of students who had delegated this choice to their family or school.

We found a significant difference in perceived utility and satisfaction [$F(1,63) = 23.6$, $p = 0.000$] with an effect size of $\eta^2 = 0.23$. Indeed, profitability (Figure 1) and satisfaction (Figure 2) appear to be higher in those who have made a personal choice, which participants indicated as having been guided by curiosity, rather than in those who have chosen based on the suggestions of the school [$F(1,63) = 0.705$, $p = 0.404$, $\eta^2 = 0.009$] or their family [$F(1,63) = 0.008$, $p = 0.931$, $\eta^2 = 0.000$].

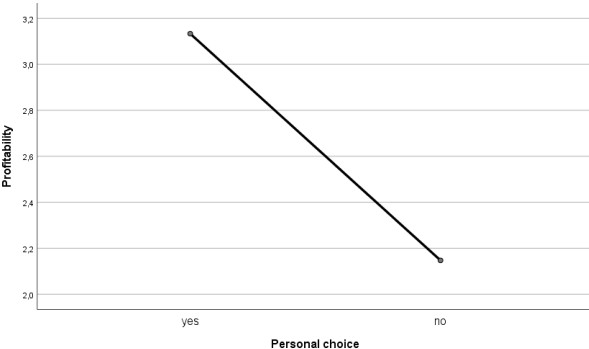

**Figure 1.** ANOVA profitability x personal choice.

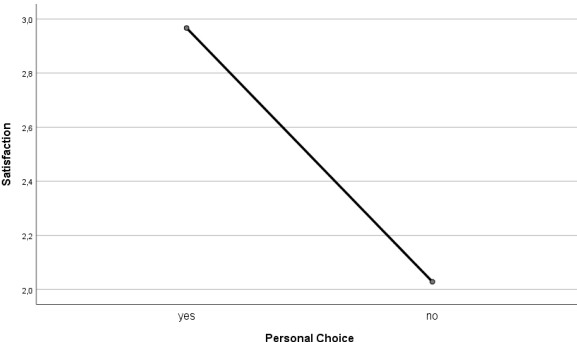

**Figure 2.** ANOVA satisfaction and x personal choice.

The autonomous choice can be seen as an indicator of greater involvement and motivation on the part of students, as they are driven by their intrinsic curiosity and desire to explore the world of work. Conversely, those who follow external suggestions may perceive the experience as less personalized or congruent with their own interests.

*5.2. Qualitative Results*

First and foremost, what is most noticeable upon initial examination of the material is the stereotypical and gaunt way young people have conveyed their opinions, it was possible to define two main clusters: negative and positive, with related codes.

Starting from the negative issue, three main codes were detected: useless (1), confusion (2), and bad experience (3).

About the first code, numerous participants responded to the question "Do you think that the SWA experience has been useful for the future university or career choices you will have to make?", by simply stating that "it was useless, because it had nothing to do with the studies undertaken over the years" (R. 11). This result suggests a perceived mismatch between the SWA experience and the knowledge acquired through their years of study. This observation raises questions about the effectiveness and alignment of the SWA program with participants' educational and career aspirations.

Moreover, the recurring assertion of a "waste of time" prominently emerges from the interviews. Indeed, several interviewees described their SWA experience using these words: "precious time taken away from our studies" (R. 56; R. 43). This statement reflects a negative perception by students who are expressing a belief that the program is not adding value to their academic pursuits and is interfering with their valuable study time. The use of the attribute "precious" suggests that students highly value their time for studying and they view any divergence from this activity as a significant drawback.

This collective viewpoint raises important considerations regarding the program's alignment with the students' academic goals and the need for a more constructive and purposeful integration of work-based experiences into their educational journey.

The second negative code can be defined as "confusion": numerous complaints fall into this category of "chaos". Many students noted that "professors, students, the school, and the company did not know what to do" (R. 25), and some students, with a sense of resignation and discouragement, attribute their negative feedback directly to the poor organization and management of programs. These statements vividly express a strong and negative perception of the SWA program, participants are conveying a sense of disorder, confusion, and lack of structure. The term "chaos" suggests that the SWA experience is perceived as a disorganized and unpredictable environment, where elements seem to be in a state of upheaval or turmoil: "the company couldn't dedicate the necessary time to the project, and there was a lack of organization" (R. 33). This feedback emphasized a critical need for improved coordination and structure within the SWA programs to provide a more positive and constructive learning environment for participants. Furthermore, this lack of structure and guidance could negatively impact students' experience, diminishing their overall satisfaction with the program.

The third code represents the most severe instance, as students report a bad experience. Some participants felt "exploited" by the companies where they were supposed to learn and train "they had us make photocopies, rearrange, and clean vases" (R. 60). They were asked to perform specific activities not in line with the expectations, nor even with an educational purpose: "we often found ourselves cleaning or moving boxes" (R. 32). This passage underscores a very critical situation, as students report having bad experience during SWA programs. The use of the term "exploited" by participants suggests a deeply negative perception of their engagement with the partners where they were supposed to learn and train. The examples provided, such as making photocopies, rearranging, and cleaning vases, indicate that some students were assigned tasks that seemed unrelated to the intended educational goals of the program and, in general, emphasize the disconnection between the students' expectations and the actual experiences encountered during SWA. This result calls for a dual reflection, on one hand, regarding the commitment of the partners involved in the SWA programs with the educational and learning goals of the program itself, and on the other hand, concerning the students' limited awareness and knowledge of the working world. Both of these aspects will be extensively addressed and discussed in the subsequent discussions.

The second main cluster identified represents the perceived positive dimensions of SWA. Through the analysis, three sub-clusters or codes were described: vocational meaning (1), learning (2), and hands-on experience (3).

With regard to the vocational purpose of the SWA experiences (1), it appears that many students had the opportunity to clarify their ideas about interests or future decisions thanks to SWA. This function was also observed in cases where the experience was not appreciated by students: "it allowed me to understand that I will never do that kind of job because I don't like it" (R. 45).

"It has opened up new opportunities, helping me understand if it was the subject, I wanted to continue studying through the laboratory tests I faced" (R. 27); these observations confirm the power of SWA's self-orientation aim in regard to vocational choices. The following quotation detects a sense of agency about their professional future: "I believe it has been very useful because, besides introducing me to other realities, it has also clarified my ideas about what I want to do in my future" (R. 07). All these passages emphasize how SWA, if adequately performed, serves as a valuable tool for self-orientation in vocational choices, providing practical insights that aid students in making informed decisions about their academic and professional future.

The second code, about the learning experience (2), was mostly related to soft skills:

"I have been able to improve my autonomous research and problem-solving skills" (R. 18) or "I have perfected my collaboration and communication skills" (R. 42). The focus on soft skills learning is profoundly significant because cognitive and relational abilities, manifesting as behaviors, are key competences that can be implemented in various situations and contexts on a daily basis. According to these students' statements, they perceive to have learned both skills related to self-perception (such as, problem solving or time management) and skills related to external relations with others (such as, communication and collaboration). Furthermore, time management and goal setting are relevant soft skills, which very often are honed after the high school experience, only when students face working or university environments. "I have learned to manage responsibilities, time and reach work goals" (R. 77). The acquisition of soft skills not only enhances individual employability but also contributes to personal and professional development, laying a foundation for success in future academic and career pursuits.

The third code refers to the hands-on experience (3), a vital aspect of the SWA program. The statement "It has brought me closer to the world of work and has allowed me to gain hands-on experience with important software" (R. 67) underlines that direct engagement in practical activities provides students with a tangible connection to the working world, offering them the opportunity to apply theoretical knowledge gained in the classroom. Moreover, the possibility to directly experience some kind of practical activity is very useful to enhance agency and employability perception, nurturing the understanding of career exploration as well. Hands-on experience, therefore, serves as a powerful bridge between theoretical learning and practical application, enriching students' understanding of career exploration and preparing them for the demands of future academic and professional environments.

### 5.3. T-Lab Results

The contents of interviews were also analyzed with a quantitative methodology using the statistical software T-Lab, capable of returning a mapping of the characterizing contents. An automatic analysis of the content was performed, which starts from the idea that the more we refer to specific language families in our speech, the more these concepts are active in our mind.

The first thing T-Lab allows to do with textual material is to analyze word occurrences and co-occurrences. The output of the software shows in the middle the most cited word, and all around, the words that co-occur the most with it, according to an association index: the Cosine coefficient. In graphical terms, the more two words co-occur, the more they are closed in the dimensional space [60]. It is always possible to "dialog" with the software and ask to put in the middle a specific word of interest for the user to have a graphical representation of its associations; in such a sense, T-Lab can assist the user in following both an automatic analysis path and a customized one.

Moreover, T-Lab allows you to obtain the phrase where the two words co-occur, and this cue is particularly useful in terms of a mix method, because with just a click, you can access the original textual material that can be analyzed by discourse analysis. We checked occurrences and co-occurrences, setting a frequency threshold of four. As Figure 3 shows, the value association of the thematic elements is graphically represented in terms of distance from the keyword in the centre.

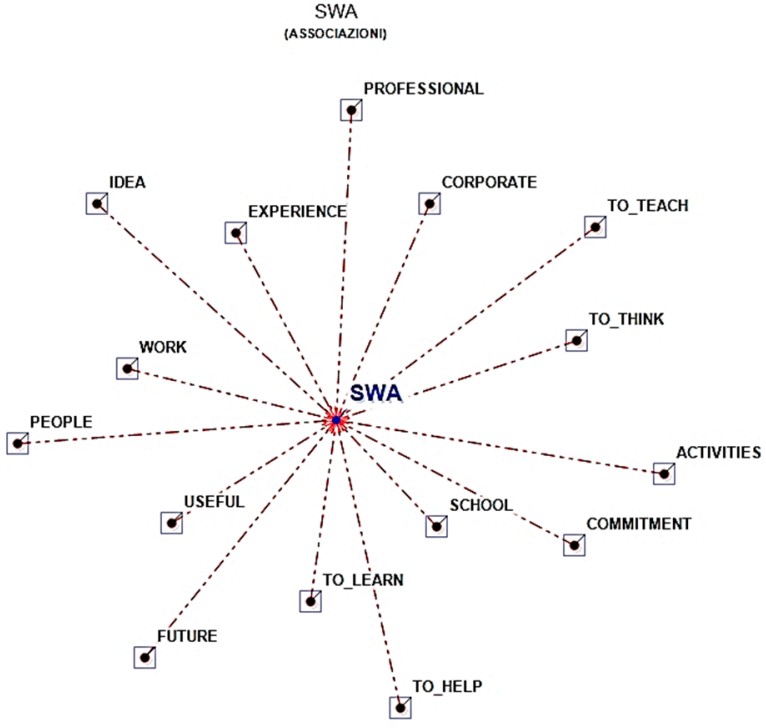

**Figure 3.** Graphical associations with the lemma SWA.

The most frequently occurring lemma in the interview was the target lemma "School-to-work Alternation (SWA) (71 occurrences).

As we can see in Table 3, the acronym SWA is strongly associated with the lemma "experience" (Cosin, 0.26), as well as with the verbs "to teach" (Cosin, 0.23) and "to learn" (Cosin, 0.22). The emphasis on the association between SWA and the concept of "experience" is noteworthy, highlighting the integral role it plays in the respondents' understanding. Moreover, the strong correlation with the verbs "to teach" and "to learn" underscores the educational aspects of SWA, suggesting that it is perceived not only as a practical experience but also as a dynamic learning opportunity. These correlations confirm the results that emerged from qualitative analysis, particularly with regard to the positive aspects of SWA's activities. Respondents, in particular, view SWA as a "useful" (Cosin, 0.20) experience capable of bridging typically distinct realms, such as "work" (Cosin 0.42), "Corporate" (Cosin 0.36), and "School" (Cosin 0.59).

The second most commonly mentioned lemma is "useful" (56 occurrences) (Figure 4 and Table 4).

It appears to be strongly related to the lemma "commitment" (Cosin 0.46), "satisfaction" (Cosin 0.36), and "Interesting" (Cosin 0.30). According to quantitative results, this pattern indicates a significant connection between the perceived usefulness of SWA and key factors such as commitment, satisfaction, and interest. Respondents seem to consistently acknowledge and underscore the practical value of the SWA, particularly when the "choice" (Cosin 0.38) of SWA was a personal one.

Moreover, it is also associated with the verb "to help" (Cosin 0.29) and with the lemmas "future" (Cosin 0.32), "university" (Cosin 0.33), and "work" (Cosin 0.39). These correlations are particularly important in terms of academic and career guidance, suggesting

a connection between the perceived usefulness of the school-to-work alternation and its role in assisting students. The ties to concepts like the future, university, and work underscore its relevance in shaping educational and vocational paths, emphasizing its potential to provide valuable support and direction in these crucial aspects of students' lives.

**Table 3.** Coefficient of Cosin and Chi² of co-occurrence with lemma School–Work Alternance (SWA).

| LEMMA_B | COEFF | CE_B | CE_AB | CHI2 | (*p*) |
|---|---|---|---|---|---|
| School | 0.593 | 25 | 25 | 22.586 | 0 |
| To_learn | 0.462 | 29 | 21 | 3.160 | 0.075 |
| Useful | 0.459 | 56 | 29 | 1.748 | 0.186 |
| Work | 0.421 | 35 | 21 | 0.065 | 0.798 |
| Experience | 0.407 | 34 | 20 | 0.007 | 0.93 |
| Corporate | 0.363 | 18 | 13 | 1.707 | 0.191 |
| To_think | 0.356 | 16 | 12 | 2.137 | 0.144 |
| Commitment | 0.332 | 25 | 14 | 0.062 | 0.803 |
| To_help | 0.287 | 17 | 10 | 0.003 | 0.955 |
| Future | 0.285 | 14 | 9 | 0.241 | 0.623 |
| Professional | 0.274 | 12 | 8 | 0.392 | 0.531 |
| To_teach | 0.274 | 12 | 8 | 0.392 | 0.531 |

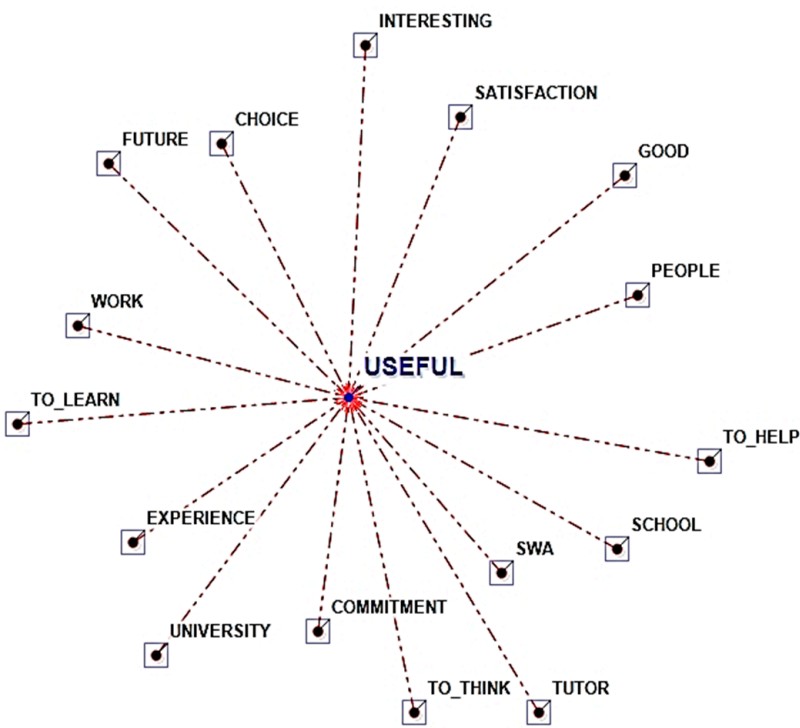

**Figure 4.** Co-occurrences with the lemma USEFUL.

**Table 4.** Coefficient of Cosin and Chi² of co-occurrence with lemma USEFUL.

| LEMMA_B | COEFF | CE_B | CE_AB | CHI2 | (*p*) |
|---|---|---|---|---|---|
| Swa | 0.459 | 71 | 29 | 1.748 | 0.186 |
| Commitment | 0.454 | 25 | 17 | 6.183 | 0.013 |
| Experience | 0.412 | 34 | 18 | 0.940 | 0.332 |
| Work | 0.383 | 35 | 17 | 0.140 | 0.,707 |
| Choice | 0.379 | 15 | 11 | 5.182 | 0.023 |
| Satisfaction | 0.356 | 9 | 8 | 7.230 | 0.007 |
| People | 0.349 | 21 | 12 | 1.290 | 0.256 |
| School | 0.347 | 25 | 13 | 0.470 | 0.493 |
| To_think | 0.334 | 16 | 10 | 2.043 | 0.153 |
| University | 0.334 | 16 | 10 | 2.043 | 0.153 |
| To_learn | 0.322 | 29 | 13 | 0.017 | 0.894 |
| Future | 0.321 | 14 | 9 | 2.152 | 0.142 |
| Interesting | 0.306 | 19 | 10 | 0.410 | 0.522 |
| To_help | 0.291 | 17 | 9 | 0.394 | 0.53 |

Finally, we conducted a personalized analysis, instructing the software to map co-occurrence with the word "experience" (Figure 5).

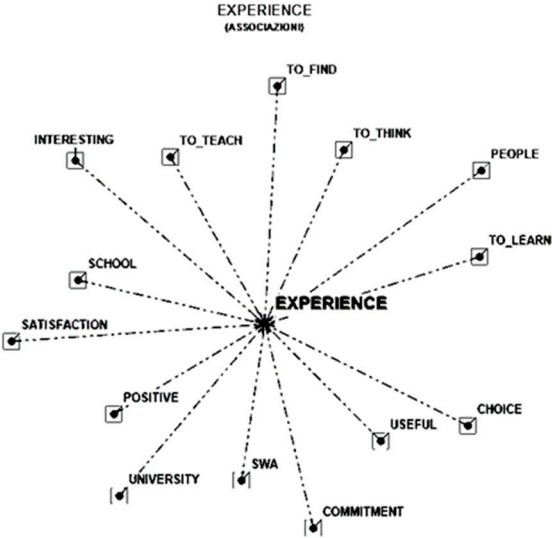

**Figure 5.** Co-occurrences with the lemma experience.

This choice is grounded in the primary objective of our study, which is to delve into students' experience with School-to-work Alternation (SWA) and unravel the intricacies of their interactions within both working and educational settings. The lemma "Experience" (Table 5) is strongly associated with the verbs "to teach" (Cosin 0.34), "to learn" (Cosin 0.31), and "to find" (Cosin 0.27) and with the lemmas "people" (Cosin 0.26), "commitment" (Cosin 0.31), and "university" (0.30). This result confirms again the multifaceted nature of SWA, indicating its connection not only to educational processes, like teaching and learning, but also to broader concepts such as interpersonal relationships and commitment. In particular, the link with "university" suggests that SWA could play a pivotal role in preparing students for the academic challenges and lifestyle changes associated with higher education. So, according to qualitative results, the SWA program appears to serve as a

bridge, facilitating a smoother transition from the structured educational environment of school to the more independent and diverse academic setting of university life.

**Table 5.** Coefficient of Cosin and Chi² of co-occurrence with lemma EXPERIENCE.

| LEMMA_B | COEFF | CE_B | CE_AB | CHI2 | (*p*) |
|---|---|---|---|---|---|
| Useful | 0.412 | 56 | 18 | 0.940 | 0.332 |
| Swa | 0.407 | 71 | 20 | 0.007 | 0.93 |
| Positive | 0.400 | 9 | 7 | 12.040 | 0.001 |
| School | 0.377 | 25 | 11 | 4.070 | 0.044 |
| To_teach | 0.346 | 12 | 7 | 6.144 | 0.013 |
| To_think | 0.342 | 16 | 8 | 4.486 | 0.034 |
| To_learn | 0.318 | 29 | 10 | 0.827 | 0.363 |
| Choice | 0.309 | 15 | 7 | 3.006 | 0.083 |
| Commitment | 0.308 | 25 | 9 | 1.034 | 0.309 |
| University | 0.300 | 16 | 7 | 2.310 | 0.129 |
| Satisfaction | 0.285 | 9 | 5 | 3.705 | 0.054 |
| Interesting | 0.275 | 19 | 7 | 0.901 | 0.342 |
| To_find | 0.271 | 10 | 5 | 2.654 | 0.103 |
| People | 0.261 | 21 | 7 | 0.376 | 0.539 |

Furthermore, the adjectives "interesting" (Cosin 0.27), "Useful" (Cosin 0.41), and "positive" (0.40) seem to suggest diverse dimensions of the students' experiences with SWA. These descriptors imply that students perceive the SWA program not only as intellectually stimulating but also as a valuable and affirmative aspect of their educational and vocational career. The positive connotations of these adjectives further underscore the potential benefits and impact of SWA, particularly in shaping a positive trajectory for students entering the university phase of their education.

## 6. Discussion

The present study delved into the multifaceted impact of School–Work Alternance (SWA) on students' experience, with a particular focus on their perceptions regarding employability, sense of agency, and self-orientation. The findings reveal a positive consensus among participants, underscoring the significant enhancement of employability attributed to the integration of work experience within the academic curriculum. However, the study addresses the flip side of the coin, identifying some threats to the successful implementation of School–Work Alternation programs.

First of all, the need to align theoretical knowledge with real-world applications emerges as a crucial aspect, able to foster a deeper understanding of the field and impart practical skills, strategic for success in the competitive job market. However, this alignment cannot simply translate into a mere juxtaposition of practical activities alongside typical theoretical lessons. Instead, it must manifest as a cohesive translation of theoretical knowledge into practical experience capable of conveying a sense of utility in everyday life. This nuanced integration is vital for ensuring that the theoretical foundations taught in classrooms are not only comprehended but also practically applied, enhancing the overall value and relevance of the SWA experience for students.

Secondly, the need to improve the coordination and organization of SWA programs becomes apparent, underscoring the importance of establishing a well-structured and organized framework for SWA initiatives. Results highlight a potential disconnect between various stakeholders, including professors, students, schools, and companies. This lack of coordination can lead to inefficiencies, unclear responsibilities, and an overall disorganized

learning environment for participants. So, addressing this aspect is paramount to ensure a smoother and more effective execution of SWA programs. This means implementing clear communication channels, defining roles and responsibilities, and establishing streamlined processes to a more positive and constructive learning experience. This last concept becomes fundamental also from another perspective: company tutors should explain the activities to students, framing even those less stimulating tasks or those that may seem seemingly distant from a learning objective in the correct way (i.e., moving boxes or cleaning). This emphasis on the importance of communication could be included in the agreement between the school and the company, thus making the stakeholders involved in the process aware of it. Furthermore, it is noteworthy to acknowledge that the discontent expressed by certain students regarding the undertaken path may, to some extent, stem from their limited exposure to the working world. As this represents the inaugural work experience for many of them, they might find themselves unfamiliar with fundamental concepts like the commitment required and the notion of constructive effort. This lack of prior familiarity with professional expectations could contribute to the dissatisfaction experienced during the SWA program.

From a practical point of view, to address the issue of dissatisfaction and unprofitability perceived by several students, various measures could be implemented. Firstly, schools could offer orientation and initial training sessions before the start of the SWA program, to explain expectations, dynamics, and fundamental concepts such as commitment and dedication. Furthermore, schools could provide individual counseling to students, helping them to better comprehend the opportunities and challenges in the working world and discuss personal expectations, career goals, and professional development. In parallel, a crucial role could be played by partners involved in the SWA program, which could organize informative meetings, company visits, and orientation sessions at the workplace to give students a clearer picture of the expectations in the working world. Simultaneously, in addition to technical skills, schools and partners could focus on enhancing soft skills such as stress management, effective communication, and resilience, fundamental for successfully navigating daily challenges. Finally, regularly collecting feedback from students during and after the SWA experience could be a significant step. This ongoing assessment process would allow schools and partners to evaluate the program's effectiveness and make any necessary improvements. This perspective probably explains the replacement of SWA with the "Path for Transversal Skills and Orientation", with the aim to provide students with the opportunity to acquire a broader range of knowledge, including aspects related to professional orientation and transversal skills.

This study has revealed not only the challenges and areas of dissatisfaction associated with SWA experience but has also highlighted the extremely positive aspects experienced by the students. These positive outcomes testify to the significant impact that the SWA program can have on students' personal and professional development. In the following paragraphs, we discuss some of these positive elements, contributing to a more comprehensive understanding of the effectiveness and value of this educational pathway. In particular, this study shed light on the role of SWA in empowering students to actively shape their educational and career trajectories, thanks to the development of a proactive sense of agency. In the following paragraph, all of these results will be discussed point by point.

First of all, the analysis of participants' perceptions regarding the impact of SWA on employability revealed that the integration of work experience within the academic curriculum significantly enhanced their employability. In particular, the alignment of academic knowledge with real-world applications was highlighted as a key contributor to improved employability. Several studies have consistently shown that practical experience gained through programs like SWA positively influences students' readiness for the workplace [65,66]. The integration of work seems to develop a deeper understanding of their field and apply theoretical knowledge in real-world scenarios. Moreover, the hands-on nature of SWA provided students with practical skills directly applicable to the workplace, making them more competitive candidates in the job market. This experiential learning

approach not only enhances their technical proficiency [67,68] but also fosters the development of soft skills such as communication [69], teamwork [70], and problem solving [71]. This result suggests several practical implications, first in terms of recruitment and talent management. Employers often seek candidates who not only possess academic knowledge, but also demonstrate the ability to navigate professional environments and contribute effectively to workplace tasks. Furthermore, the positive consensus among participants regarding the impact of SWA on employability suggests that these programs can play a crucial role in preparing students for the transition from academia to the workforce [51,52]. In particular, the alignment of academic learning with practical experiences could create future workers who are better equipped to meet the demands of today's dynamic job market [53,72].

Secondly, participants consistently reported that SWA played a pivotal role in the development of a sense of agency. Thus, engaging in work experiences while pursuing academic studies empowered students to take control of their educational and career trajectories actively. According to the literature, it appears that the combination of academic studies and work experiences, as exemplified by SWA, has a significant impact on the development of a sense of agency [73,74]. In fact, by engaging in work experiences alongside school studies, students seem to gain a sense of control over their actions. This empowerment may arise from the dual exposure to theoretical knowledge, derived from academic contests, and practical skills acquired through work experiences. Thus, according to the literature, our results seem to suggest the idea that combining work and academic pursuits positively influences individuals' perception of agency and contributes to a proactive approach in managing educational and career paths [75], reinforcing their belief in their ability to influence and shape their professional future.

Thirdly, the results shed light on the role that SWA can play in enhancing the self-orientation of students. It seems that the ability to actively navigate their career paths and their perception regarding the program's utility are really valuable in assisting them in their future vocational choice. According to our sample, the results of the ANOVA can confirm how positive the impact of self-determination is: when students perform their own choice without school or family influences, they also are more satisfied with the SWA experience. The past literature has already investigated the role of contextual influences on students' well-being in making informed career or academic choices [76,77].

This study has some limitations as well: the sample is quite small and not representative of the entire population. In order to gain a sound understanding of the SWA experience, it would be essential to gather a more extensive and varied sample. Additionally, the study's cross-sectional design poses limitations to our ability to establish causal relationships or monitor changes over an extended period.

Practical implications can be hypothesized for several stakeholders: students, organizations, and educational institutions. With regard to students, the importance of a self-oriented and self-determined choice can be confirmed, as well as the subsequent perceived satisfaction with the SWA experience. Young adolescents should trust their curiosity rather than follow parents' or teachers' expectations; this assumption is crucial for the efficacy of SWA in terms of vocational guidance and empowerment. Moreover, students who participate in SWA experience develop job-specific skills and gain real-world experience, so that they become more attractive as future job candidates. Both for students and organizations, networking opportunities are a valuable outcome of the SWA experience; according to our participants, some students appreciate the importance of building connections, potentially leading to future job opportunities.

The results of the study also suggest practical implications for organizations; first and foremost, an efficient SWA experience is a crucial factor for the employer branding issue. Participants' statements raised concerns regarding corporate responsibility for their vocational role: it is essential to provide a real hands-on experience to students and make them really involved in the work context. Organizations may benefit from the opportunity of pre-recruiting or a pipeline of skilled and motivated young students.

Practical implications for educational institutions tackle the gap between the theoretical and practical world of work, so that schools can better design and adapt the curricula, ensuring that students are equipped with basic soft skills needed to make an informed and aware choice.

In conclusion, this study offers a nuanced perspective on the School–Work Alternation (SWA) experience, unveiling both challenges and remarkable positive aspects encountered by participating students. So, addressing the identified challenges and capitalizing on the positive aspects can contribute to optimizing the overall effectiveness and value of this valuable educational pathway.

**Author Contributions:** Conceptualization, M.C.; methodology, M.C.; formal analysis, S.F. and T.G.; data curation, T.D.F. and S.I.; writing—original draft, S.F. and T.G.; writing—review and editing, S.F. and T.G.; supervision, M.C.; project administration, M.C. All authors have read and agreed to the published version of the manuscript.

**Funding:** This research received no external funding.

**Institutional Review Board Statement:** Ethical review and approval were waived for this study as it was conducted according to the guidelines of the Declaration of Helsinki and the guidelines of the Italian Psychological Association. Since no treatments, demands or manipulations even potentially causing physical, psychological, or social discomfort to participants were carried out, ethical review and approval of the study was not required, in accordance with national and local legislation.

**Informed Consent Statement:** Informed consent was obtained from all subjects involved in the study.

**Data Availability Statement:** Data available on request to the author.

**Conflicts of Interest:** The authors declare no conflicts of interest.

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
