# Peer review of "Bridging the Gap between Theoretical Learning and Practical Application: A Qualitative Study in the Italian Educational Context"

_education, doi:10.3390/educsci14020198_

Round 1

Reviewer 1 Report

Comments and Suggestions for Authors

Reviewer’s suggestions and comments on the Manuscript entitled:

Bridging the gap between theoretical learning and practical application: a qualitative study in the Italian Educational Context

Manuscript ID: education-2784783

The authors have to incorporate the time that students spend in their working places. How long, and how frequently?

Did the students have the opportunity to change working places, or did they have the experience from just one type of job?

Was the personnel professionally prepared and motivated to welcome and work with young students?

Instead of using the decimal comma use decimal dot.

There were 63 students interviewed on their experience regarding School-Work Alternation (SWA). I would expect some more clear analysis, for example, how many of them were 100 % satisfied or dissatisfied, if such results are obtained. Average grades of the Variable are relatively bad, somewhere around 3, this has to be commented, on since grade 3 isn’t something that it has to be proud of.

Negative issues have to be analyzed more since expressions like “it was useless, because it had nothing to do with the studies undertaken over the years”, “waste of time” “precious time taken away from our studies”, “confusion”, “professors, students, the school, and the company did not know what to do” “the company couldn’t dedicate the necessary time to the project, and there was a lack of organization”, “exploited”, “we often found ourselves cleaning or moving boxes” “they had us make photocopies, rearrange, and clean vases” are hard accusations, we have to admit.

However, students are very young, and without experience, and the main question is what they can do. Do we have time to give them jobs that are interesting with a higher sense of responsibility? Moving boxes, cleaning of working place, or making photocopies is something that we all have to do from time to time, especially when we are in junior positions. This is okay by me if it is combined with inspiring work. If it was all bad then SWA program has strong weaknesses regarding the targeted company, if not then we have to ask ourselves selves, are the students mature enough to understand that work isn’t just fun and joy? For example, being a part of the scientific staff in some experiments is interesting, but washing laboratory dishes afterward is not, but that is a part of the job. I supposed that the authors have all possible examples and I would expect a clearer analysis on this rather complex problem.

This manuscript has high citation potency. However, I expect some reasonable answers regarding my criticism. The authors have to understand that this is a positive criticism. Therefore I recommend to the Editorial Office to reconsider this manuscript after major revision.

Author Response

Dear reviewer,

thank you for your in-depth observations. We appreciate your insightful review of our study. Your feedback has been precious in guiding our revisions, particularly in enhancing the analysis of negative results and criticisms raised by participants.

We have now implemented a brief but comprehensive description of SWA regulations in Italy, detailing the duration and the kind of experience. Regarding the negative issues highlighted, we acknowledge the importance of analyzing these aspects more comprehensively.

We also had the chance to add the very interesting point that you raised, about the significance of job tasks; thus we commented in the discussion section, on the importance of communication between company tutors and young students regarding the meaning of activities.

Additionally, we have addressed issues of confusion and lack of organization within the program. We have discussed the need for clearer communication channels and additional support for both students and partners involved in SWA.

Thank you for your positive and constructive criticism, which we greatly appreciate. We have carefully considered your suggestions and believe we addressed your concerns by providing additional clarification in the text, especially in the discussion section.

Best regards,

the authors

Reviewer 2 Report

Comments and Suggestions for Authors

Authors have submitted a manuscript titled “Bridging the gap between theoretical learning and practical application: a qualitative study in the Italian Educational Context”. The authors have aimed to qualitatively research students' SWA experience.

1.       Introduction and problem statement is clearly written. One note: this idea seems to appear twice in the text: “SWA is embedded within the national educational system 61 and regulated by the Ministry of Education, Universities and Research (MIUR). The program typically involves students alternating between traditional classroom settings and real-world workplaces,” in section 1 and 2.1. Check if you can rephrase it or leave only in one place.

2.       A brief description of Italian education system in general would be useful for the foreign readers, e.g., at what age does SWA start and what level of education is it in Italy?

3.       Two main theoretical concepts “Employability” and “Sense of Agency” are chosen by authors for exploration. Their meaning in the context of SWA is well-justified.

4.       In the Abstract it is mentioned that qualitative analysis was used, but later on in the text it also appears that “mix-methods” were applied, and a part of the results are clearly calculated by quantitative analysis methods, including an evaluation of items on a Likert scale. Thus, this should be explained clearer.

5.       Table 2 was not visible in the Manuscript version that was available.

6.       In tables it would be advised to write full names of the scales or measures to make it more clear to the reader. For example, in Table 5: what does “To_find” mean? Also please check if the guidelines for manuscript defines how many decimals to leave after a comma in the table. Some have two, some have more numbers (e.g., “0,007612   ;   0,93”). Similar in Tables 4 and 3. I would advise to use two decimals everywhere (e.g., 0.27).

7.       I would suggest moving the Table 1 into the Results section, as usually the descriptive statistics is presented at the beginning of the Results section. Otherwise, it is illogical to present results and only then introduce data analysis approach.

8.       In the Results / Discussion section it is noted about the negative views of SWA by the students, if this is something important, it should be also mentioned in the Abstract.

9.       Please check the formatting of the References list. It looks like some of the text is in different font size or different font type.

Author Response

Dear reviewer,

 Thank you for your observations. We appreciate your insightful review of our study.

We enriched the introduction section with a comprehensive description of the Italian education system and more details about the SWA procedure. We also could improve the communication in the abstract (about both the applied method and the negative results). Thank you for all the precious annotations regarding the organization of text and tables, we hope tables can now be more clear with the full name of the scales and with two decimals. For what concerns the example in table 5 “To_Find”: from tables 3 to 5 there are the T-Lab results, in the first column report the cited words in co-occurrence with the target word. Lemma can be verbs, adjectives, or nouns, in case of two separate words (such as for the infinite form: “to find or to teach”) the T-Lab software needs an underscore between them, otherwise, it does not allow the analyses.

Thank you again for your suggestions and feedback.

Reviewer 3 Report

Comments and Suggestions for Authors

Thank you for your work. It is undoubtedly of interest to investigate the effects of school-to-work alternance (which has long since ceased to be so called also from a regulatory point of view).
The contribution is presented as research on school-to-work alternance (now PCTO) proposing generalisations from a sample of 63 subjects about whose choice no information is given.
The description of the process and methodology appears unclear and there are significant elements of internal inconsistency.
The literature supporting both the theoretical framework and the conclusions is in many cases not very relevant to what is claimed. The article needs a comprehensive review.

Author Response

Dear reviewer,

thank you for taking the time to review our manuscript on the experience of School-to Work Alternance, now known as PCTO. We appreciate your thoughtful feedback and constructive suggestions, which we believe significantly contributed to the improvement of our work.

We acknowledge your concern regarding the clarity of the process and methodology description. So, we have revised these sections to enhance their clarity and coherence. Specific attention has been given to addressing any ambiguity and ensuring a more transparent presentation of our research approach.

Your suggestion for a comprehensive review is duly noted, and we refined our manuscript accordingly. 

Thank you again for your time and valuable input.

Round 2

Reviewer 1 Report

Comments and Suggestions for Authors

The authors did a good job in revising the manuscript in line with the concerns of the reviewers.

Reviewer 3 Report

Comments and Suggestions for Authors

The article has been clarified and can be accepted.